# Data linkage in medical research

Katie Harron

**bmj**medicine
Correspondence to: Dr Katie Harron, UCL Great Ormond Street Institute of Child Health, Population Policy and Practice, London, UK; k.harron@ucl.ac.uk

Data linkage provides an opportunity to harness existing data for medical research. This article outlines key approaches for data linkage, and describes methods used to quantify, interpret, and account for errors.

Data linkage combines data from different sources that relate to the same person to create a new, enhanced data resource. This technique allows researchers to exploit and enhance existing data sources without the time and cost associated with primary data collection. Linked data can be used to supplement follow-up in conventional cohort studies or trials, or to generate real world evidence by creating population level electronic cohorts that are entirely derived from administrative data (figure 1).[1 2] These longitudinal data sources help us to answer questions that require large sample sizes (eg, for rare diseases) or whole population coverage (eg, for pandemic response planning), which consider a wide range of risk factors and outcomes (including social determinants) and are especially powerful for capturing populations that are hard to reach.[3 4] Figure 1 illustrates two real world examples of how data linkage has been used to inform medical research, to improve clinical trial follow-up and to examine outcomes from birth.

## Choosing the right approach

A barrier to generating linked data that are fit for purpose is the availability of accurate identifiers that can be used to link the same person across multiple data sources.[5] Recording of unique identifiers such as the NHS number nearly always involves some degree of error or missing data.[6] Therefore, linkage often depends on the use of non-unique identifiers such as name, postcode, and date of birth, or even indirect identifiers such as procedure dates or other clinical variables.[7] In combination, these variables can allow us to identify records that belong to the same person—but errors, changes over time, or missing data can still hamper attempts to find the correct link.

Linkage methods have two main categories: deterministic (rule based) methods and probabilistic methods involving match weights or scores. For example, national hospital records in England (Hospital Episode Statistics) are linked longitudinally for the same person using a three step deterministic algorithm that looks for exact agreement on a combination of identifiers: NHS number, date of birth, postcode, and sex.[8] Probabilistic linkage assigns a weight to each record pair, representing the likelihood that two records belong to the same individual, given the agreement or not between identifiers. In effect, probabilistic match weights allow all possible deterministic rules for a set of available identifiers to be ranked.[9] A threshold weight is then used to classify records as links or non-links.

## Linkage error

Irrespective of the linkage methods implemented, use of imperfect and dynamic identifiers can lead to linkage error. Linkage errors manifest as false matches (where records belonging to different individuals are linked together) or missed matches (where records belonging to the same individual are not linked). Analogous to false positives and false negatives, these linkage errors can be viewed through a diagnostic accuracy lens (table 1). While carefully designed linkage algorithms and high quality recording of identifying information can facilitate accurate linkage, even small amounts of error can lead to bias.[10] This problem is particularly evident when individuals from certain subgroups are less likely to link accurately.[11] For example, maintaining consistent linkage quality across ethnic groups can be a challenge.[12]

Any linkage strategy will allow, to a certain extent, a trade-off between the two types of errors.[9] In probabilistic linkage, this trade-off depends on the choice of threshold (that is, the weight above which records have been classified as links; figure 2). As the threshold is lowered, sensitivity of linkage (that is, the proportion of true matches captured) increases, but the false match rate also increases. A similar diagram could be drawn to represent the trade-off in deterministic linkage as we decide which matching rule or match rank should be used to classify records. Sensitivity analyses can be used to explore the impact of the choice of threshold or matching rule on results.[13]

Design of a linkage strategy should be informed by the intended application or research question. For example, when creating a system to support drug administration using linked records, we would need to ensure that treatments are not delivered to the wrong patient: a conservative or specific approach aiming to minimise false matches would be appropriate. Conversely, use of linked data to invite members of the public for screening programmes might prioritise coverage at the expense of sending some invitations in error: a more sensitive approach might be appropriate in this setting. Minimising the

## KEY MESSAGES

⇒ Data linkage in medical research allows researchers to exploit and enhance existing data sources without the time and cost associated with primary data collection

⇒ Methods used to quantify, interpret, and account for errors in the linkage process are needed, alongside guidelines for transparent reporting

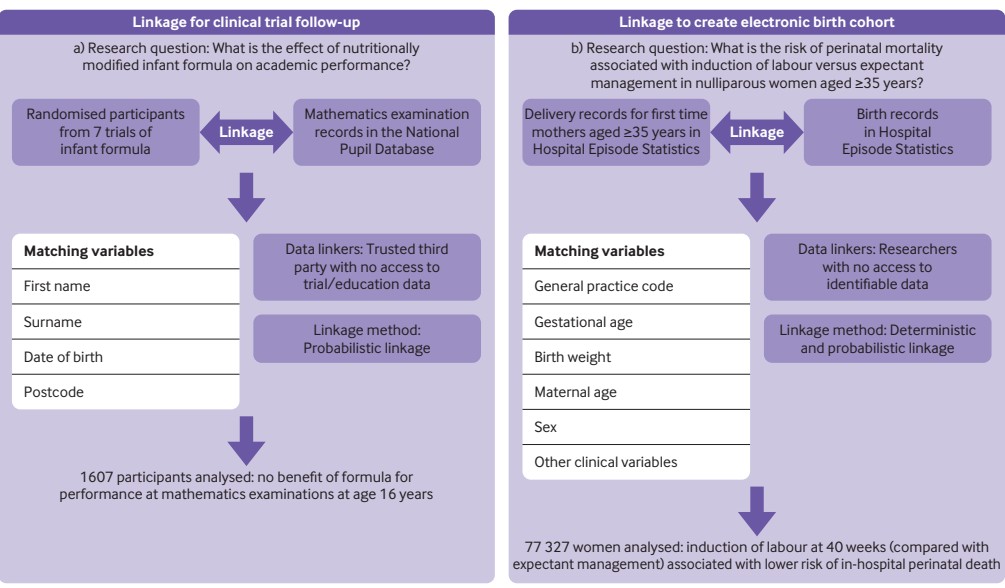

Figure 1 | Examples of linkage used to support clinical trials and create whole population cohorts.[21][22]

difference between error types might also important in some situations. For example, when mortality rates are estimated by linking a cohort to mortality records, the correct rate might still be estimated if the number of false and missed matches cancel out.

## Quality control and accounting for linkage error

Several methods can be used to evaluate the quality of linkage.[14] These methods focus on identifying potential sources of bias (that is, which characteristics are associated with errors) by examining the characteristics of records that are linked versus unlinked, or that have high versus low quality identifier data, or that are easily identifiable as having been linked incorrectly (eg, through quality control checks).[15] Accounting for linkage error in analysis is an ongoing area of methodological research, but includes approaches that view uncertainty in linkage as a missing data problem best handled with some form of multiple imputation or weighting, and those that attempt to quantify and adjust for errors using quantitative bias analysis.[16] Reporting guidelines are available that explicitly aim to support transparent reporting of linkage studies.[5][17]

## Remaining challenges

The biggest barriers to realising the full potential of data linkage as a powerful research tool are gaining and maintaining public trust, and reducing the costs, delays, and inefficiencies in how linked data are made available for research in the public interest.[18][19] For example, proposals to routinely link health records in primary and secondary care in order to support planning and research in England (from care.data in 2012 to General Practice Data for Planning and Research in 2021) have repeatedly raised public concerns about the lack of transparency surrounding how linked data are to be used, processes for opting out, and commercial interests. However, the covid-19 pandemic has highlighted that efficient and secure access to linked data can support agile and responsive research: building on the success of initiatives such as OpenSafely and the

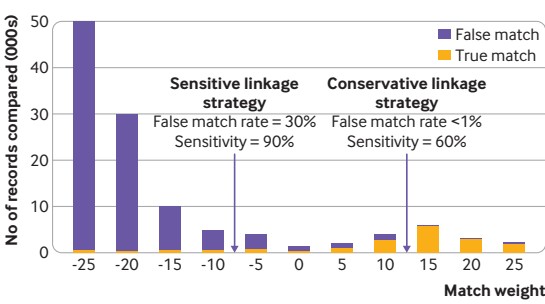

Figure 2 | Example of trade-off between false matches and missed matches in probabilistic linkage. In this example, probabilistic match weights are used to classify records as belonging to the same individual or not. A threshold of ≥15 would mean that <1% of linked records were false matches but 40% of the true matches were not captured. Decreasing the threshold to −5 would increase the proportion of true matches captured to 90%, but would also increase the false match rate to 30%

### Table 1 | Linkage accuracy tool

| Assigned link status | True match status | |
| --- | --- | --- |
| | Match (pair from same individual) | Non-match (pair from different individuals) |
| Link | True match a | False match b |
| Non-link | Missed match c | True non-match d |

Sensitivity (or recall)=a/(a+c); specificity=d/(b+d); positive predictive value (or precision)=a/(a+b); negative predictive value=d/(c+d). In practice, the number of non-matches will usually far outweigh the number of matches, and so the positive predictive value and sensitivity are more informative than the specificity and negative predictive value.

British Heart Foundation's CVD-COVID-UK consortium (both of which link primary and secondary care data for the UK) could provide a way forward.[1 20]

**Contributors** KH wrote the article.

**Competing interests** We have read and understood the BMJ policy on declaration of interests and declare the following interests: none.

**Patient and public involvement** Patients and the public were not involved in the design, or conduct, or reporting, or dissemination plans of this research.

**Provenance and peer review** Commissioned; not externally peer reviewed.

**Data availability statement** No data are available.

**ORCID iD**
Katie Harron http://orcid.org/0000-0002-3418-2856

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
