## [Reviewer comments · BMJ Medicine]

ARTICLE DETAILS

TITLE (PROVISIONAL)	Data linkage in medical research
AUTHORS	Harron, Katie

VERSION 1 - REVIEW

REVIEWER	Reviewer 1: Riley, Richard , Keele University, School of Medicine, No conflict of interest
REVIEW RETURNED	19-Dec-2021

GENERAL COMMENTS	Thank you for the opportunity to review this article for BMJ Medicine, which I enjoyed reading and is well-written, and will provide an excellent RMR article. As such, I only have generally minor comments, as follows. 1) The opening line “Methods to quantify, interpret and account for errors, and guidelines to support transparent reporting of linkage studies can facilitate important medical research, says Katie Harron” is somewhat confusing, as it sets the tone that the article is about errors – whereas the focus is on data linkage as a whole. So I would suggest to rewrite this as something like ‘Data linkage provides an opportunity to harness existing data for medical research. Here, Katie Harron outlines key approaches for data linkage and says methods to quantify, interpret and account for errors are needed, alongside guidelines for transparent reporting’ 2) In the medical literature the last five years has seen the term ‘real world evidence’ increasingly used, to represent data from the real world (i.e. not from a trial or other experiment) and populations and settings therein. I think for completeness, this article would benefit from mentioning that high quality data linkage is essential for studies that aim to provide real world evidence. I think this could even be mentioned in the very first paragraph. Here is a good thread of real-world data. https://www.ispor.org/strategic-initiatives/real-world-evidence 3) In the first paragraph, it is good to see mention of advantages such as for rare diseases and increasing sample sizes. The examples in Box 1 are also excellent. To draw the reader in and link the text and this box better, I would finish paragraph 1 by saying something more explicit like ‘Box 1 gives two real examples of how data linkage has been used to inform medical research, one to improve clinical trial follow-up and another to examine outcomes from birth.’ 4) End of first page says “In probabilistic linkage, this trade-off depends on the choice of threshold, ... “ – define that you mean by threshold (i.e. the probability above which you are happy to define that records can be linked).
---

	5) When a threshold is used, should the subsequent statistical analyses be repeated following different choices of thresholds for the linkage? i.e. sensitivity analyses are needed? This seems to be never done in practice, but after reading this excellent piece, I am now inclined to always want to see this done. Please comment. 6) "... or that are easily identifiable as having been linked incorrectly (either through internal or external validation" – what do you mean by internal and external validation here? This has different connotations in different fields, so briefly add clarity (how do you show something is 'valid'? And what is internal?) 7) The sub-heading "Quality control" is better "Quality control and accounting for linkage error" 8) The sentence "Methods to quantify, interpret and account for errors, and guidelines to support transparent reporting of linkage studies, can facilitate important medical research" is rather vague and covers many topics – and is repeating the current introductory statement to the piece. Be more specific. How can these methods support medical research? What guidelines are available for reporting? Any particular recommendations for the reader? Etc 9) The section "Choosing the right approach and managing error" is quite large relative to the other sections, and I suggest splitting it into two, with 'Types of approaches to data linkage' being the first two paragraphs and "Managing error" being the other parts. Just a suggestion to make the piece flow better. 10) I saw a complimentary piece from the author here (https://www.gov.uk/government/publications/joined-up-data-in-government-the-future-of-data-linking-methods/quality-assessment-in-data-linkage) , which I think should be cited, as the reader can then get more details than space allows in this RMR. I hope these comments are helpful. Some of my suggestions may add slightly to the word count, but I suggest that the BMJ Medicine Editors allow this to enable a more complete and informative piece, as I would not want the author to remove any text.
--	--

VERSION 1 – AUTHOR RESPONSE

Reviewer comment	Author response
1) The opening line "Methods to quantify, interpret and account for errors, and guidelines to support transparent reporting of linkage studies can facilitate important medical research, says Katie Harron" is somewhat confusing, as it sets the tone that the article is about errors – whereas the focus is on data linkage as a whole. So I would suggest to rewrite this as something like 'Data linkage provides an opportunity to harness existing data for medical research. Here, Katie Harron outlines key approaches for data linkage and says methods to quantify, interpret and account for errors are needed, alongside guidelines for transparent reporting"	Thank you, suggestion taken.

2) In the medical literature the last five years has seen the term 'real world evidence' increasingly used, to represent data from the real world (i.e. not from a trial or other experiment) and populations and settings therein. I think for completeness, this article would benefit from mentioning that high quality data linkage is essential for studies that aim to provide real world evidence. I think this could even be mentioned in the very first paragraph. Here is a good thread of real-world data.	Agreed this is helpful. I have now included this in the first paragraph.
3) In the first paragraph, it is good to see mention of advantages such as for rare diseases and increasing sample sizes. The examples in Box 1 are also excellent. To draw the reader in and link the text and this box better, I would finish paragraph 1 by saying something more explicit like 'Box 1 gives two real examples of how data linkage has been used to inform medical research, one to improve clinical trial follow-up and another to examine outcomes from birth.'	Suggestion taken.
4) End of first page says "In probabilistic linkage, this trade-off depends on the choice of threshold, ... " – define that you mean by threshold (i.e. the probability above which you are happy to define that records can be linked).	Altered to "i.e. the weight above which you have classified records as links"
5) When a threshold is used, should the subsequent statistical analyses be repeated following different choices of thresholds for the linkage? i.e. sensitivity analyses are needed? This seems to be never done in practice, but after reading this excellent piece, I am now inclined to always want to see this done. Please comment.	This is a good point and an area of debate! Whilst some recommend that sensitivity analyses are helpful in understanding the direction and extent of potential bias from linkage error, others say that there will be some error with any threshold, and that sensitivity analyses may still not give you the correct range of answers. With this caveat, I have included the following sentence: "Sensitivity analyses can be used to explore the impact of the choice of threshold or matching rule on results". There are examples of where this has been done in the literature, and I have now included a reference to demonstrate this.
6) "... or that are easily identifiable as having been linked incorrectly (either through internal or external validation" – what do you mean by internal and external validation here? This has different connotations in different fields, so briefly add clarity (how do you show something is 'valid'? And what is internal?)	To avoid confusion, I have altered this to "through quality control checks". The cited reference has more details.
7) The sub-heading "Quality control" is better "Quality control and accounting for linkage error"	Changed as suggested.

8) The sentence “Methods to quantify, interpret and account for errors, and guidelines to support transparent reporting of linkage studies, can facilitate important medical research” is rather vague and covers many topics – and is repeating the current introductory statement to the piece. Be more specific. How can these methods support medical research? What guidelines are available for reporting? Any particular recommendations for the reader? Etc	I agree this was vague and have included a more direct statement with references to the appropriate guidelines: “Reporting guidelines are available that explicitly aim to support transparent reporting of linkage studies”.
9) The section “Choosing the right approach and managing error” is quite large relative to the other sections, and I suggest splitting it into two, with ‘Types of approaches to data linkage’ being the first two paragraphs and “Managing error” being the other parts. Just a suggestion to make the piece flow better.	Suggestion taken.
10) I saw a complimentary piece from the author here, which I think should be cited, as the reader can then get more details than space allows in this RMR.	I have now included this reference under the Quality control section.